# Long-Tailed Pygmy Rice Rats Modify Their Behavioural Response and Faecal Corticosterone Metabolites in Response to Culpeo Fox but Not to Lesser Grison

**DOI:** 10.3390/ani11113036

**Published:** 2021-10-22

**Authors:** María del Carmen Hernández, André V. Rubio, Isabel Barja

**Affiliations:** 1Departamento de Biología, Campus Universitario de Cantoblanco, Universidad Autónoma de Madrid, C/Darwin 2, 28049 Madrid, Spain; isabel.barja@uam.es; 2Departamento de Ciencias Biológicas Animales, Facultad de Ciencias Veterinarias y Pecuarias, Universidad de Chile, Santa Rosa 11735, La Pintana, Santiago 8820808, Chile; 3Centro de Investigación en Biodiversidad y Cambio Global (CIBC-UAM), Campus Universitario de Cantoblanco, Universidad Autónoma de Madrid, C/Darwin 2, 28049 Madrid, Spain

**Keywords:** predator cues, predation risk, *Oligoryzomys longicaudatus*, *Lycalopex culpaeus*, *Galictis cuja*, glucocorticoids, corticosterone

## Abstract

**Simple Summary:**

Prey species must fine-tune their antipredator responses to survive, but also to achieve a positive net energy balance which will enhance biological fitness. Given this, we investigated if *Oligoryzomys longicaudatus* would adapt their behavioural and physiological antipredator responses depending on their relative presence in the predator’s diet. By exposing this rodent species to culpeo fox and lesser grison faeces, we found that *O. longicaudatus* behavioural and physiological strategies were modulated depending on the predator’s diet. Specifically, rodents would trigger their antipredator responses in the presence of the most dangerous predator, the culpeo, which consumes a higher proportion of *O. longicaudatus* compared to the lesser grison. Our findings could be of importance for the development of more humane and efficient strategies to control rodent populations.

**Abstract:**

Even though behavioural and physiological reactions to predation risk exhibited by prey species have received considerable attention in scientific journals, there are still many questions still unsolved. Our aim was to broaden the knowledge on one specific question: do long-tailed pygmy rice rats adapt their behavioural and physiological antipredator strategies depending on the predator species? For this question, we live-trapped in a temperate forest in Southern Chile long-tailed pygmy rice rats (*Oligoryzomys longicaudatus*), which were exposed to three predator odour phases (Phase 0: preliminary, no predator cues; Phase 1: one plot with culpeo fox faeces (*Lycalopex culpaeus*), one plot with lesser grison (*Galictis cuja*) faeces and one plot acting as a control with no odour; Phase 2: post treatment, no predator cues). We measured the behavioural response by the capture ratio. To assess the physiological stress response, we collected fresh faecal samples to quantify faecal corticosterone metabolites (FCM). Our results showed that *O. longicaudatus* increased both the capture ratio and FCM levels in the presence of culpeo cues. Culpeo foxes have higher densities in the study area than *G. cuja* and exhibit a higher activity pattern overlap with *O. longicaudatus.* Moreover, it has been also been reported in other regions that *L. culpaeus* consumption of *O. longicaudatus* is more frequent compared to *G. cuja* diet. The increase in capturability could be because traps can be regarded as a shelter in high-risk settings, but it can also be explained by the predator inspection behaviour. The increase in FCM concentrations during culpeo treatment can be linked to the adaptive mobilisation of energy to execute antipredator responses to increase survival chances.

## 1. Introduction

Predation risk drives the evolution of numerous prey adaptations [1,2]. There exists a vast variety of responses; in particular, behavioural and physiological adaptations are key to cope with predator pressure [3,4,5]. Among rodents, the most common behavioural responses to predation risk are linked to changes in daily activity patterns, use of space, social behaviour and feeding habits [6,7,8]. In particular, previous studies have demonstrated that the presence of direct predator cues can prompt individuals to avoid those areas because the perceived predation risk is higher [3,9,10]. However, previous field studies using live trapping to measure predator avoidance have also found that rodents could consider traps as shelter in the presence of predator cues [11].

The physiological stress response is a highly refined neuroendocrine-systemic pathway that plays a major role in the ability of animals to overcome changes in the environment [12,13], including predation risk. In particular, it is a crucial process to increase the energy available to the individual [14] to escape from predators. When the prey detects the presence of the predator, the stimulus triggers the activation of the hypothalamic–pituitary–adrenal axis (HPA) axis, increasing the secretion and release of glucocorticoids (GC) by the adrenal gland [15] and leading to the mobilization of reserves needed to deal with the predatory event [12,16]. Whereas the short-term release of GC is an adaptive response which enhances the probability of surviving by redirecting energy from non-vital processes [14,17], a sustained release of GC can produce severe deleterious effects (such as immune suppression and endocrine disruption), leading to a critical reduction in biological fitness [12,14,18].

Despite predation risk assessment in prey species having been widely addressed in the literature, numerous aspects of these adaptive strategies remain unclear. For example, how predator species which exhibit marked differences in density, activity patterns and diet can affect the antipredator strategies of such prey species.

In this study, we analysed the behavioural and physiological responses of *Oligoryzomys longicaudatus* to two predators which show different activity patterns, densities and dietary preferences. On the one hand, the culpeo fox seems to be more abundant in the study area [19] and their daily activity patterns are cathemeral, exhibiting intermittent activity throughout the 24 h cycle [19]. In other regions, the culpeo fox has been described as a facultative trophic specialist mainly feeding on rodents and lagomorphs, depending on local abundances [20,21]. *O. longicaudatus* can be an importance source of food for this species in Chilean ecosystems [22,23] as high densities of this prey can be found in central and southern temperate forests [24,25]. On the other hand, the lesser grison appears to be scarce in the study area compared to fox abundance [19]. This species primarily feeds on rodents and lagomorphs [26,27], but *O. longicaudatus* do not represent an important fraction of their diet in central and southern Chile [26,28,29]. Moreover, lesser grisons have been reported to be mainly diurnal and crepuscular [30,31] and camera trapping records confirmed that lesser grisons in the area are mostly diurnal [19], while *O. longicaudatus* is chiefly considered nocturnal [32,33,34].

The aim of this study was to evaluate the behavioural and physiological stress responses (through capture ratios and faecal corticosterone metabolite levels, respectively) of *O. longicaudatus* to two different predators (culpeo and lesser grison) that differ in their daily activity patterns, abundance and diet. According to the risk allocation hypothesis [35], we expected that *O. longicaudatus* would increase their antipredator efforts when perceived predation risk was higher (i.e., in the presence of culpeo cues, because they pose a greater threat for *O. longicaudatus* than the lesser grison). Therefore, we predicted a lower capture ratio and an increased GC release in rodents exposed to culpeo cues.

## 2. Materials and Methods

### 2.1. Study Area

The study was conducted in a temperate forest located in Huelemolle, at the Villarrica lake basin (39°16′ S, 71°48′ W), Araucanía Region (southern Chile). The climate in this area is temperate-humid with a short dry season (<4 months) in summer (January–March) and an average rainfall of 2000 mm distributed throughout the year. Minimum and maximum temperatures range from 10.4 °C to 25.3°, respectively, in the warmest month (January) and 4.2 °C to 12.1 °C in the coldest month (July). The vegetation comprises forests dominated by Patagonian oak (*Lophozonia obliqua*) and coigue (*Nothofagus dombeyi*), mainly associated with Chilean laurel (*Laurelia sempervirens*), olivillo (*Aextoxicon punctatum*), ulmo (*Eucryphia* cordifolia) and lingue (*Persea lingue*) (*Gajardo* 1993). Carnivore species found in the area include puma (*Puma concolor*), chilla fox (*Lycalopex griseus*), molina’s hog-nosed skunk (*Conepatus chinga*), güiña (*Leopardus* guigna), culpeo fox (*Lycalopex culpaeus*) and lesser grison (*Galictis cuja*) [36,37]. Rodent species found in the study area include *O. longicaudatus* (71.5%), *Abrothrix longipilis* (11.4%), *Rattus rattus* (10.6%) and *Abrothrix olivaceus* (6.5%), during autumn and winter [A. Rubio, unpublished data].

### 2.2. Experimental Design

Live-trapping was performed during the austral winter (June 2019), when *O. longicaudatus* reaches peak abundances in temperate forests [38,39]. The study area was divided into three plots (A/B/C) 200 m apart from each other in order to avoid pseudoreplication (i.e., capturing the same mouse in two different plots). In each plot, we placed 42 Sherman-like traps (240 × 80 × 90 mm; Schulz Instruments, Santiago, Chile) in a 6 × 7 grid, with 5 m of distance among them [40,41]. Traps were activated at dusk and checked daily at dawn. All traps were placed under vegetation to buffer extreme environmental conditions.

The experiment was divided into three different and consecutive phases: phase 0 (preliminary), phase 1 (treatment) and phase 2 (post treatment). Each phase took place over 3 consecutive days. During phase 0, no carnivore faecal odour was placed in any plot in order to determine the basal behavioural and physiological responses of *O. longicaudatus*. In phase 1, one plot (A) was used as a control with no experimentally added carnivore faecal odour and the other two plots were subjected to carnivore faecal odour: one plot (B) with culpeo fox faeces (*L. culpaeus*) and one plot (C) with lesser grison (*G. cuja*) faeces. In this phase, 10 g of faecal material (see section simulation of predation risk by faecal odour) was placed outside each trap. Finally, in phase 2, we removed the faecal material on the first day, in order to evaluate the effect of the decrease in predation risk over time. All traps during the three study phases were baited with rolled oats with vanilla essence.

### 2.3. Simulation of Predation Risk by Faecal Odour

Culpeo fox and lesser grison faeces were used to simulate predation risk because they are two common *O. longicaudatus* predators [23,41]. In general, both fox and mustelid faeces and urine have been demonstrated to successfully trigger antipredatory responses in wild rodents [10,11,42,43]. Faeces were gathered from captive adult animals (2 males and 2 females for both culpeo and lesser grison) from the Metropolitan Zoo (Santiago, Chile). All animals were on a carnivorous diet and we only collected fresh scats, i.e., only those with a layer of mucus, an elevated level of hydration and strong odour [44,45]. All faeces samples were frozen at −20 °C until treatment preparation. Due to the fact that volatile compounds can vary depending on the season or individual factors [46,47,48], all faeces were mixed to provide a homogeneous stimulus across all treated traps. Faecal material was replaced every day at sunset to ensure correct odour effectiveness.

### 2.4. Data Collection

Each captured individual was identified to the species level based on external morphology. Sex was determined using the anal–genital distance, which is longer in males than in females [49]. In the same way, reproductively active females were classified on the basis of the presence of prominent nipples and perforated vaginal membranes, whereas reproductive active males were identified due to the increased size of their testicles, that usually descend into the scrotal sac [49]. Body mass was measured employing a 50 g hand-held scale (PESOLA, 50 g). All captured animals were marked on specific body areas (paws, inner ear area, tail) with harmless waterproof paints to identify possible recaptures in each phase and to avoid pseudoreplication. After handling, animals were released at their place of capture.

### 2.5. Faeces Collection and Quantification of Faecal Corticosterone Metabolites

Quantifying faecal cortisol/corticosterone metabolites has proven to be a reliable non-invasive method to measure GC levels in wild species [8,11,50,51,52]. Since circadian rhythms can produce differences in excretion patterns [53], we gathered faecal samples each day at the same time in the mornings (between 8:00–10:00 a.m.). Faeces contaminated with urine were not collected to avoid any possible cross contamination [53]. In the field, faecal samples were stored in Eppendorf tubes in a portable cooler with wet ice (4 °C), later (around 12:00 a.m.), after data collection, all samples were stored at −20 °C. Faecal glucocorticoid extraction was performed following the protocol of Barja et al. [50] and Navarro-Castilla et al. [43]. First, frozen faecal samples were dried in the laboratory oven at 90 °C for 4 h. Then, 0.05 g of each sample was weighed and mixed it with 500 μL of 80% methanol and 500 μL of phosphate buffer. Eppendorfs were vortexed by hand for 15 s and then multivortexed for 16 h. After that, the samples were centrifuged for 15 min at 2500× *g*. FCM quantification was performed using a commercial corticosterone enzyme immunoassay kit (DEMEDITEC Diagnostics GmbH, Kiel, Germany; corticosterone: DEMEDITEC D24145) to determine the faecal metabolite hormone levels by the same procedure published in previous studies [10,11,43]. The corticosterone kit was validated by determining the parallelism, precision and accuracy: Parallel displacement curves were obtained by comparing serial dilutions (1:32, 1:16, 1:8, 1:4, 1:2, 1:1) of pooled faecal extracts with the standard curves, resulting in both curves being parallel. Precision was tested through intra- and inter-assay coefficients of variation for faecal samples, and the corticosterone intra-assay coefficient of variation was 8.7% and inter-assay was 9.3%. The mean recovery (accuracy) percentage from the assayed hormone was 92.7%. Results are expressed as nanograms of corticosterone metabolites per gram of dry faeces matter.

The long-tailed pygmy rice rat is one of the main reservoirs of Hantavirus in Chile, which entails a high risk of transmission when animals are kept in captivity. Therefore, the authors did not perform an ACTH challenge to avoid unnecessary health risks, and since the required permits to keep this species in the laboratory would only be granted under strictly justified conditions. In addition, the physiological validation clearly supports that the kit used was correctly measuring corticosterone levels in the faecal samples. Considering that during the treatment all individuals were exposed during 3 days to the same continuous stressor (predators’ faeces), we consider that the ACTH was not strictly required. Furthermore, this minimizes unnecessary animal suffering and, thus, we carried out a non-invasive procedure.

### 2.6. Statistical Analysis

To analyse the behavioural response of *O. longicaudatus* to predation risk, we conducted a generalized linear model (GLM) using capture frequency (capture vs no capture) as a response variable in a model with binary distribution and logit as the link function. We included as explanatory variables the predation risk treatment (Phase 0 (Preliminary)/Phase 1 (Control)/Phase 1 (Culpeo)/Phase 1 (Lesser grison)/Phase 2 (Post treatment), and the plot (A/B/C). Even though the selected plots were similar in vegetation composition and percentage of canopy cover (A. Rubio, unpublished data), we wanted to control for its possible effects.

We conducted a generalized linear mixed model (GLMM) with gaussian distribution and identity link to analyse the physiological stress response of *O. longicaudatus*. Log-transformed FCM levels were set as the response variable for the model. Explanatory variables considered in the model were the predator treatment (Phase 0 (Preliminary)/Phase 1 (Control)/Phase 1 (Culpeo)/Phase 1 (Lesser grison)/Phase 2 (Post treatment) and the sex (female/male). The individual was set as a random effect to control for its possible effect. Model selection was based on Akaike Information Criterion (AIC).

Sex and reproductive status have been previously described as being able to modulate rodents’ physiological stress response to predation risk [8,10,11,43]. To control for its possible effects, we included sex as a fixed effect in the models. Reproductive status was not considered in the analyses because all individuals captured were not breeding. For both models, post hoc comparison was performed using the Mann–Whitney U test.

The software used to perform the statistical analysis were SPSS 23.0 for Windows (SPSS Inc., Chicago, IL, USA) and R 4.1.1 (http://www.r-project.org) for the GLMM. Data are represented as mean ± standard error (SE). Results were considered significant at α < 0.05.

## 3. Results

### 3.1. Behavioural Response: Capturability

The total number of *O. longicaudatus* captures in this study was 299, corresponding to 88 different individuals. Since trapping efforts were not the same for each treatment, we corrected each number of captures for the corresponding trapping effort in traps-night (Figure 1). A total of 121 of the captures were made under preliminary treatment (total trapping effort for control traps was 378), 24 captures corresponded to control treatment (trapping effort was 126), 46 corresponded to culpeo treatment (trapping effort 126), 34 corresponded to lesser grison treatment (trapping effort 126), and 74 to the post treatment (trapping effort 378). Results of the GLM analysing the capture ratio (Table 1), showed that there were statistically significant differences in capture probability depending on the predation risk treatment (Figure 1), with the culpeo treatment being the one with the highest capturability. Post hoc testing with the Mann–Whitney U test revealed statistically significant differences between preliminary and post treatment (*p* < 0.005) and culpeo fox and post treatment (*p* < 0.0001).

### 3.2. Physiological Stress Response

Results of the GLMM analysing the individual and external factors which modulated *O. longicaudatus* FCM levels are shown in Table 2. The predation risk treatment was the only explanatory variable responsible for the variation in the FCM levels. The highest mean FCM levels were found during Phase 1 culpeo treatment (Figure 2), followed by Phase 1 controls. The lowest FCM concentrations were detected in *O. longicaudatus* captured during Phase 0 (preliminary), followed by Phase 1 lesser grison treatment and Phase 2 (post treatment). The Mann–Whitney U post hoc test demonstrated that differences between groups were found between the culpeo and preliminary treatments (*p* < 0.05), culpeo and post treatment (*p* < 0.05) and between the culpeo and lesser grison treatments (*p* < 0.05).

## 4. Discussion

### 4.1. Behavioural Response

It has been described that predator faeces can cause an avoidance effect in many small mammal species (e.g., *Apodemus sylvaticus, Microtus agrestis, Myodes glareolus, Arvicola amphibius*, *Oryctolagus cuniculus* and so forth) [10,43,54,55,56]. However, in our study, *O. longicaudatus* capturability increased during the culpeo fox treatment while the lowest capture ratio was found during the control treatment and the post treatment. Previous studies have also suggested that rodents can regard traps as a suitable shelter under increased risk of predation [11], as the benefits of finding a safe spot to hide outweigh the risks taken by exploring these unknown devices. Given this, *O. longicaudatus* may have used traps as a refuge during the culpeo treatment (which could explain the increase in rodent capturability), as it was the most dangerous scenario compared to the other ones. Another possible explanation would be the “predator inspection” phenomenon, which suggests that prey approach the predator cues to extract information and to improve their risk assessment [57,58,59].

Chemically-mediated predator avoidance may be modulated by the predator’s diet, as volatile composition of predator by-products would be different depending on the food ingested [5,60,61,62,63]. In this study, behavioural antipredator response to lesser grison treatment could have not been triggered because lesser grison is not one of the main predators of *O. longicaudatus*; thus, the perceived risk was lower, so it could be more optimal to not divert so much time and energy from other vital activities [35].

Moreover, previous studies have suggested that faeces of carnivores that defecate in latrines could provide less valuable information for predation risk assessment, as their scats are only concentrated in certain spots of their territory [8,54,64,65]; thus, they do not accurately reflect the predator’s presence or movement patterns. Since lesser grisons defecate in latrines [29,66], it might be possible that the uncertain benefits of triggering the antipredator responses in this setting would not counterbalance the costs for *O. longicaudatus*.

In the case of the post treatment, capturability could have been lower because rodents were not under an immediate predator threat, so they were not compelled to seek shelter. In the first phase, capturability was higher than the post treatment, which could be because traps were novel objects and *O. longicaudatus* could have been exploring these devices to assess their risks [67,68,69]. *O. longicaudatus* is considered a scansorial species [70] but it frequently forages on the ground and explores novel sources of food [71]. They possess a remarkable jumping ability and partial bipedalism, which allow them to escape quickly from predators when they are foraging on the ground [72,73].

The post-hoc test was unable to detect statistically significant differences involving the control treatment, probably due to the small sample size of this treatment compared to the preliminary and post treatment.

### 4.2. Physiological Stress Response

Despite several studies failing to corroborate the activation of the HPA axis of rodent species under predation risk [8,43,74], our results were in accordance with studies that reported a significant glucocorticoid release in the presence of predator cues [10,11,75].

We found that HPA axis activation of *O. longicaudatus* was higher during culpeo treatment. As we expected, rodents triggered the glucocorticoid release when facing the most threatening treatment (i.e., culpeo cues), which would allow prey to mobilize energy towards the organs and tissues where it is needed to display successful antipredator strategies (e.g., flight, finding shelter, defensive attack), and thus, survive [76,77]. Even though rodents could have felt safe inside traps, once they were caught, they were continuously exposed to fox faeces (which were next to the entrance of the trap), and this may have triggered and maintained a higher physiological antipredator response [10,11]. As for the lesser grison treatment, it seems that the threat it posed was not enough to activate the HPA axis. As we mentioned before, lesser grison is scarce in the study area [19], and they have different active periods (*O. longicaudatus* is nocturnal while lesser grisons are mostly diurnal) [19]. Moreover, studies conducted in other regions showed that *O. longicaudatus* is not an important part of the lesser grison diet, [26,28,29]. Therefore, the costs of displaying this physiological antipredator strategy could surpass the possible benefits obtained, and hence, it would not be adaptive to increase GC production. Since there are no studies on the diet of mesocarnivores in the area, future studies would be needed to confirm the importance of the differential consumption of the target prey species in the modulation of the antipredator responses. Additionally, the fact that lesser grisons defecate in latrines could also explain the absence of the HPA response, as the information provided in those faeces could be not sufficiently reliable [8,54,64,65]. These findings would be also explained by the threat sensitivity hypothesis [78,79], which explains that prey would assess the current risk in each different context and would differentially respond depending on the degree of perceived risk. Threat sensitivity hypothesis has been previously corroborated in small mammal species such as *A. sylvaticus* [11] and *O. cuniculus* [80].

Moreover, FCM levels decreased to baseline levels once the riskier predator treatment (i.e., culpeo cues) was removed. The physiological stress response is an energy-expensive pathway, so it appears that it would be more energy efficient and fitness-enhancing to arrest such a response once the predator has dispersed. This result would be in accordance with the optimality theory [81,82,83], which proposes that natural selection would favour certain optimal evolutionarily stable phenotypes because they outperform others, and thus would bring higher fitness to those individuals. Nonetheless, future studies focused on the analysis of this effect would be needed to confirm when exactly FCM levels drop, as lesser grison treatment was apparently unable to trigger the physiological antipredator response.

## 5. Conclusions

Taken together, our findings provide a new insight into the trade-offs involved in rodents’ risk assessment. Individuals triggered the physiological antipredator response only in the presence of the predator which shows a higher density, activity overlap and consumption of *O. longicaudatus* (i.e., culpeo fox), and thus poses a bigger threat. On the contrary, *O. longicaudatus* did not allocate energy to these strategies in the presence of predator cues of lesser grison, which is mainly diurnal, not abundant in the area and does not regularly consume *O. longicaudatus*. Moreover, following energy optimization principles, the physiological stress response of *O. longicaudatus* faded to baseline levels after the encounter with the most dangerous predator. These results could be of critical importance not only in contributing to a better understanding of animal ecophysiology, behaviour and the adaptive strategies displayed by prey but also for the management and conservation of wildlife populations (e.g., rodent control strategies compatible with biodiversity conservation).

## Figures and Tables

**Figure 1 animals-11-03036-f001:**
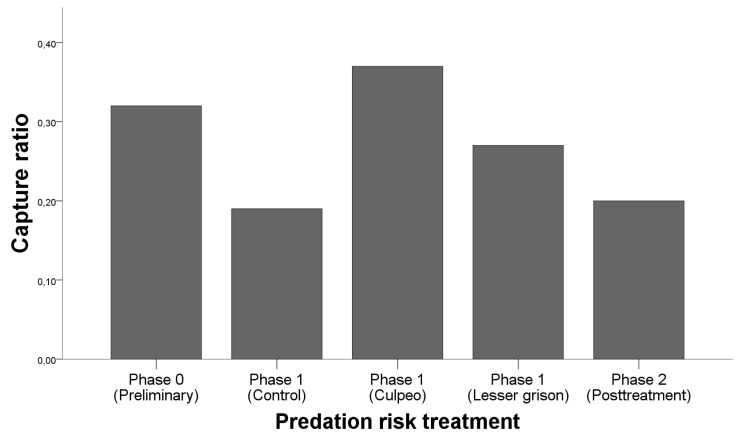
Capture ratio (number of captures of *O. longicaudatus* corrected by the trapping effort in traps-night) depending on the predation risk treatment (Phase 0—Preliminary/Phase 1—Control/Phase 1—Culpeo/Phase 1—Lesser grison/Phase 2—Post treatment).

**Figure 2 animals-11-03036-f002:**
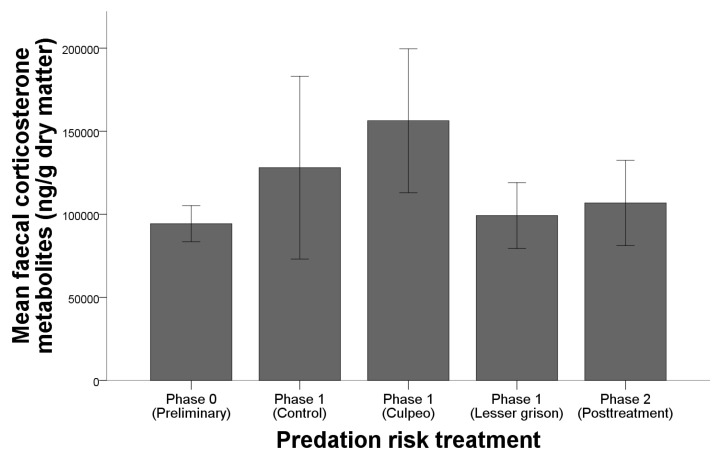
Mean faecal corticosterone metabolite levels of *O. longicaudatus* depending on the predation risk treatment (Phase 0—Preliminary/Phase 1—Control/Phase 1—Culpeo/Phase 1—Lesser grison/Phase 2—Post treatment).

**Table 1 animals-11-03036-t001:** Results of the GLM, analysing the effects of experimental and environmental factors on *O. longicaudatus* capturability.

Effect	*F*	df	*p*-Value
Intercept	160.70	1	<0.001
Predation risk treatment	18.94	4	<0.001
Plot	3.75	2	1.153

**Table 2 animals-11-03036-t002:** Results of the GLMM, analysing the effect of experimental and environmental factors on *O. longicaudatus* faecal corticosterone metabolite levels.

Effect	Estimate	Std. Error	df	*t* Value	*p*-Value
Intercept	11.25	0.07	103.90	152.39	<0.001
Treatment—Control	0.10	0.11	247.98	0.88	0.3817
Treatment—Culpeo	0.38	0.10	262.49	3.65	<0.001
Treatment—Lesser grison	0.04	0.11	262.05	0.39	0.699
Treatment—Posttreatment	0.04	0.09	249.97	0.47	0.639
Sex—Male	0.11	0.07	57.24	1.53	0.132

Random factor (Individual): variance: 0.0059; std dev: 0.0766.

## Data Availability

Not applicable.

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
