# Peer review of "Long-Tailed Pygmy Rice Rats Modify Their Behavioural Response and Faecal Corticosterone Metabolites in Response to Culpeo Fox but Not to Lesser Grison"

_animals, 2021, doi:10.3390/ani11113036_

Round 1
Reviewer 1 Report
In this study authors aim to assess the behavioural and physiological response to predator risk by the long-tailed pygmy rice rat Oligoryzomys longicaudatus. Specifically, authors investigated if long-tailed pygmy rice rat would adapt its behavioural and physiological antipredator strategies (in terms of capturability and faecal corticosterone metabolites, respectively) depending on its proportional presence in two different predators’ diet, i.e. culpeo fox Lycalopex culpaeus and lesser grison Galictis cujadiffering. For this purpose, authors live-trapped rats, which were exposed to culpeo fox and lesser grison faeces. Rats’ fresh faecal samples were then collected both in presence and absence (control phase) of predator odour to assess the physiological stress response. The behavioural response was instead measured by the rats capture ratio. Results showed that O. longicaudatus increased both capturability and faecal corticosterone metabolites in the presence of culpeo faeces, that, according to the literature, is expected to be the predator with higher consumption of O. longicaudatus in its diet. Authors claim that the increase in capturability might happen because traps are regarded as shelter in high-risk scenarios but it can also be explained by the predator inspection behaviour. Finally, the increase of faecal corticosterone metabolites in the presence of L. culpaeus faeces has been linked to the adaptive mobilisation of energy to perform antipredator responses to increase survival chances.
Overall, the manuscript is well written and the results can provide useful information on behavioural and physiological responses to predator risk by prey species. However, the study suffers from some critical issues concerning the analyses and the assumptions on which the work is based.
Please see the issues listed below:
- The assumptions of the species-specific predation risk are entirely based on a general knowledge of predators species and on a wide geographical range (see lines 71-77 and the citated literature) while no information is reported about the study area. Actually, no data on predators density and pressure on longicaudatus in the study area are reported and it is not stated if both predators actually prey in the limited live-trapping sampling area. It might be a crucial information considering that responses of the studied preys can be influenced by previous experience towards predators inhabiting the current plots. In addition, data on the predators’ local diet composition are needed, considering their adaptive and generalist food habits.
- In lines 145-147 authors point out that animals have been temporarily marked to detect possible recaptures and avoid pseudoreplications. However, it appears that authors considered all captures and recaptures in the data analyses, without an individual distinction. If the case, authors should have run a mixed model with individuals as a random effect instead. Mixed models should be used with repeated measurements and they also as make possible to take into account individual variability, which can have a strong influence in risk predator responses (as highlighted also in the citated reference, see reference 11). In addition, authors should somehow differentiate between captures and recaptures, at least in some preliminary explorative analyses, considering that capture and recapture rates might be substantially different in presence to a predator odour, when compared to a preliminary situation (i.e. no predator odour), as reported in reference 11.
- As stated in the Title, authors claim that differential consumption of long-tailed pygmy rats by culpeo fox and lesser grison cause differences in mice capturability (in addition to faecal corticosterone metabolites). Firstly, authors can not state this sentence because the differential consumption by predators has not been tested nor just specified in a detailed way. Furthermore, this difference in mice capturability does not correspond to a significant evidence. Authors should specify and discuss that post hoc tests do not show significant difference in capturability between culpeo fox and lesser grison.
- Finally, since there are pronounced inter-species differences concerning the metabolism of glucocorticoids, a biological validation would be necessary to confirm the suitability of the EIA for longicaudatus fecal samples. Hormone metabolic pathway leads to the formation of metabolites that can differ between species or even between genders of the same species. Thus, it is important to verify the reliability of hormone metabolite measurements obtained by EIA kits for each species. The authors analytically validated the EIA kit utilized but further validation is required to verify the reliability of FCM assessments: a biological or physiological validation. The authors used faeces of two predators as stress stimulus, hypothesizing that culpeo fox faeces should induce a higher FCM increase in O. longicaudatus with respect to the exposure to lesser grison faeces or in absence of any faeces. The results obtained by the authors are the ones that were expected. Thus, these findings could be considered a biological validation. However, assuming that repeated measures were used in the statistical analyses without the appropriate models, it is not possible to verify the stress response of the animal and the activity of the HPA axis. Only repeated measures analysis can demonstrate a real FCM increase in response to exposure to a predator marker. It could be important to clearly state the results of this biological validation in the discussion section. It is well known that in many species of mammals, and in particular in rodents, males and females differ for both basal and stress corticosterone levels. The authors didn’t find any sex effect on FCM levels. However, I suggest performing statistical analysis for each sex and also showing graphically the FCM levels during the three phases separately for the two genders.
Author Response
Dear reviewer,
We would like to thank you for your invaluable comments. We are confident that our manuscript has been significantly improved. Particularly, we have improved the statistical analyses, the theoretical framework, and the discussion. Please, find a below a point-by-point reply to your comments in the file attached.

Reviewer 2 Report
Editorial comments
Line 100: No hyphen for the word include
Line 113 no hyphen in the word extreme
Lines 116, 221 and Fig.1 (perhaps elsewhere) space in words post treatment
Query: Lines 103-105: are all of these rodent species ONLY available during the Autumn and Winter? Or is it true that some of these species are more seasonally available than others?
How many days between the three different phases? How many animals were captured?
It is clear that the animals were trapped but I wonder about the handling of these individuals during the data collection phase. Were the animals anesthetized-could this have influenced the results?
Also it looks like the captured animals were held somewhere to collect the faeces-describe this a bit.
Although some parts of how the captured animals were treated are discussed I found myself wondering about other important aspects (see above).
Author Response
Dear reviewer,
We would like to thank you for your helpful comments. We are confident that our manuscript has been significantly improved. Particularly, we have improved the statistical analyses, the theoretical framework, and the discussion. Please, find a point-by-point reply to your comments in the file attached.
